# Low-Resource Knowledge-Grounded Dialogue Generation

**Xueliang Zhao**[1,2]**, Wei Wu**[3]**, Chongyang Tao**[1]**, Can Xu**[3]**, Dongyan Zhao**[1,2]**, Rui Yan**[1,2,4*]

[1]Wangxuan Institute of Computer Technology, Peking University, Beijing, China
[2]Center for Data Science, AAIS, Peking University, Beijing, China
[3]Microsoft Corporation, Beijing, China
[4]Beijing Academy of Artificial Intelligence (BAAI), Beijing, China
`{xl.zhao,chongyangtao,zhaody,ruiyan}@pku.edu.cn`
`{wuwei,caxu}@microsoft.com`

## Abstract

Responding with knowledge has been recognized as an important capability for an intelligent conversational agent. Yet knowledge-grounded dialogues, as training data for learning such a response generation model, are difficult to obtain. Motivated by the challenge in practice, we consider knowledge-grounded dialogue generation under a natural assumption that only limited training examples are available. In such a low-resource setting, we devise a disentangled response decoder in order to isolate parameters that depend on knowledge-grounded dialogues from the entire generation model. By this means, the major part of the model can be learned from a large number of ungrounded dialogues and unstructured documents, while the remaining small parameters can be well fitted using the limited training examples. Evaluation results on two benchmarks indicate that with only 1/8 training data, our model can achieve the state-of-the-art performance and generalize well on out-of-domain knowledge.

## 1 Introduction

Open domain dialogue systems, due to the applications on social chatbots such as Microsoft XiaoIce (Shum et al., 2018) and virtual assistants such as Amazon Alexa (Ram et al., 2018), have drawn increasing attention from the research community of natural language processing and artificial intelligence. Thanks to the advances in neural sequence modeling (Vaswani et al., 2017; Sutskever et al., 2014) and machine learning techniques (Li et al., 2017; 2016), such systems now are able to reply with plausible responses regarding to conversation history, and thus allow an agent to have a natural conversation with humans. On the other hand, when people attempt to dive into a specific topic, they may clearly realize the gap between the conversation with a state-of-the-art system and the conversation with humans, as the system is only able to awkwardly catch up with the conversation, owing to the lack of knowledge of the subject.

We consider grounding open domain dialogue generation with knowledge which is assumed to be unstructured documents. While documents are abundant on the Web, it is difficult to obtain large scale dialogues that are naturally grounded on the documents for learning of a neural generation model. To overcome the challenge, some recent work (Zhou et al., 2018b; Dinan et al., 2019) resorts to crowd-sourcing and builds benchmarks with the source of Wikipedia. On the one hand, the datasets pave the way to the recent research on knowledge-grounded response generation/selection (Zhao et al., 2019; Lian et al., 2019; Li et al., 2019); on the other hand, we argue that there still a long way to go for application of the existing models in real scenarios, since (1) the models, especially those achieve state-of-the-art performance via sophisticated neural architectures, just overfit to the small training data (e.g., $\sim$ 18k dialogues). An evidence is that when they are applied to documents out of the domain of the training data, their performance drops dramatically, as will be seen in our experiments; and (2) it is difficult to collect enough training data for a new domain or a new language, as human effort is expensive.

---

*Corresponding author: Rui Yan (ruiyan@pku.edu.cn).

As a step towards application of knowledge-grounded dialogue generation in real-world systems, we explore how to learn a model with as few knowledge-grounded dialogues as possible, yet the model achieves state-of-the-art performance and generalizes well on out-of-domain documents. The key idea is to make parameters that rely on knowledge-grounded dialogues small and independent by disentangling the response decoder, and thus we can learn the major part of the generation model from ungrounded dialogues and plain text that are much easier to acquire. Specifically, the encoder of the generation model consists of two independent components with one for encoding the context and the other for representing the knowledge. The decoder is decomposed into conditionally independent components including a language model, a context processor, and a knowledge processor, and the three components are coordinated by a decoding manager that dynamically determines which component is activated for response prediction. The language model predicts the next word of a response based on the prior sub-sequence, and the context processor ensures coherence of the dialogue by attending over the conversation history. Both components, along with the context encoder, are independent with the extra knowledge, and thus can be pre-trained using the ungrounded dialogues. The knowledge encoder has nothing to do with dialogues, and thus can be pre-trained with the plain text. The knowledge processor is responsible for grounding response generation on the document. This part, together with the decoding manager, depends on the knowledge-grounded dialogues, but the parameters are small in size, and estimation of these parameters just requires a few training examples depending on specific domains or tasks. By fixing the pre-trained parameters, we can adapt the model to a new domain with only a little cost.

We pre-train the language model, the context processor, and the context encoder with a clean version of Reddit data (Dziri et al., 2018), pre-train the knowledge encoder using a Wikipedia dump available on ParlAI, and compare our model with baselines that hold state-of-the-art performance on two benchmarks including the Wizard of Wikipedia (Wizard) (Dinan et al., 2019) and CMU Document Grounded Conversations (CMU_DoG) (Zhou et al., 2018b). Evaluation results indicate that (1) to achieve the state-of-the-art performance, our model only needs $1/8$ training data ($\sim 2.3$k dialogues on Wizard and $\sim 0.4$k dialogues on CMU_DoG); (2) on Wizard, the model significantly outperforms the baseline models on out-of-domain documents even though the baselines have leveraged all training data, while our model is only learned with $1/16$ training data; and (3) the model performs comparably well on in-domain and out-of-domain documents in a low-resource setting.

Contributions in this work are three-fold: (1) exploration of knowledge-grounded dialogue generation under a low-resource setting; (2) proposal of pre-training the knowledge-grounded dialogue generation model with a disentangled decoder using ungrounded dialogues and documents; and (3) empirical verification of the effectiveness of the model on two benchmarks.

## 2  APPROACH

We elaborate our approach to learning a response generation model with knowledge-grounded dialogues, ungrounded dialogues, and plain text.

### 2.1  PROBLEM FORMALIZATION

Suppose that we have a dataset $\mathcal{D}_S = \{(U_i^S, D_i^S, r_i^S)\}_{i=1}^n$, where $\forall i \in \{1, \ldots, n\}$, $D_i^S$ is a document that serves as the background of the dialogue $(U_i^S, r_i^S)$, $U_i^S = (u_{i,1}^S, \ldots u_{i,n_i}^S)$ is the context of the dialogue with $u_{i,j}^S$ the $j$-th utterance, and $r_i^S$ is the response regarding to $U_i^S$ and $D_i^S$. In addition to $\mathcal{D}_S$, we further assume that there are $\mathcal{D}_P = \{D_i^P\}_{i=1}^N$ and $\mathcal{D}_C = \{(U_j^C, r_j^C)\}_{j=1}^M$ with $D_i^P$ a document and $(U_j^C, r_j^C)$ a context-response pair, $\forall i \in \{1, \ldots N\}$ and $\forall j \in \{1, \ldots, M\}$. $N \gg n$ and $M \gg n$. The goal is to learn a generation model $P(r|U, D; \theta)$ ($\theta$ denotes the parameters of the model) with $\mathcal{D} = \{\mathcal{D}_S \cup \mathcal{D}_P \cup \mathcal{D}_C\}$. Thus, given a new document $D$ with the associated dialogue context $U$, one can generate a response $r$ following $P(r|U, D; \theta)$.

Our idea is inspired by the observation on the nature of open domain dialogues: despite the fact that a dialogue is based on a document $D$, words and utterances in the dialogue are not always related to $D$ (e.g., a reply just echoing the previous turn), even for the turns from the interlocutor who has access to $D$, as demonstrated by the examples in (Dinan et al., 2019; Zhou et al., 2018b). Therefore, we postulate that formation of a response could be decomposed into three uncorrelated actions: (1) selecting a word according to what has generated to make the sentence linguistically valid (corresponding to a language model); (2) selecting a word according to the context to make the

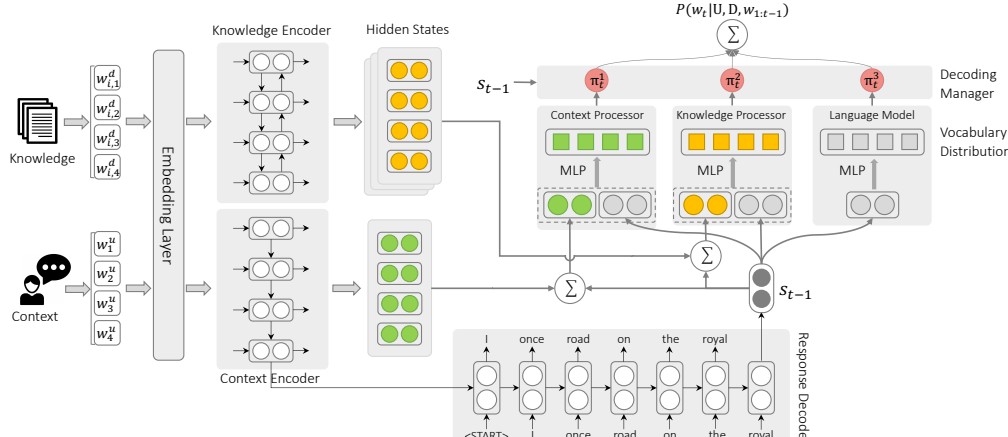

Figure 1: Architecture of the generation model.

dialogue coherent (corresponding to a context processor); and (3) selecting a word according to the extra knowledge to ground the dialogue (corresponding to a knowledge processor). The three actions can be independently learned, which becomes the key to aiding the small $\mathcal{D}_S$ with the large $\mathcal{D}_P$ and $\mathcal{D}_C$.

## 2.2 GENERATION MODEL

Figure 1 illustrates the architecture of the model. The model is made up of a context encoder, a knowledge encoder, a decoder, and a decoding manager. The major difference lies in the decoding phase which simulates the aforementioned actions by decomposing the decoder into a language model, a context processor, and a knowledge processor. The three components are independent conditioned on the hidden states of the decoder, and are coordinated by the manager.

### 2.2.1 ENCODERS

Given a dialogue context $U = (u_1, \ldots, u_l)$, the context encoder concatenates $\{u_i\}_{i=1}^l$ as $(w_1^u, \ldots, w_i^u, \ldots, w_{l_u}^u)$ with $w_i^u$ the $i$-th word in the sequence, and then exploits a recurrent neural network with gated recurrent units (GRUs) (Chung et al., 2014) to transform the word sequence into a sequence of hidden vectors given by

$$\boldsymbol{h}_1^u, \ldots, \boldsymbol{h}_i^u, \ldots, \boldsymbol{h}_{l_u}^u = \text{GRU}_{\theta_e}(\boldsymbol{e}_1^u, \ldots, \boldsymbol{e}_i^u, \ldots, \boldsymbol{e}_{l_u}^u), \tag{1}$$

where $\boldsymbol{e}_i^u$ is the embedding of $w_i^u$ initialized with GloVe (Pennington et al., 2014). $\{\boldsymbol{h}_i^u\}_{i=1}^{l_u}$ serve as the input of the context processor in decoding.

In the meanwhile, given a document $D = (d_1, \ldots, d_i, \ldots, d_m)$ with $d_i$ the $i$-th sentence, the knowledge encoder represents $d_i$ as a sequence of hidden vectors through a bidirectional GRU (Cho et al., 2014):

$$\boldsymbol{h}_{i,1}^d, \ldots, \boldsymbol{h}_{i,j}^d, \ldots, \boldsymbol{h}_{i,l_d}^d = \text{BiGRU}_{\theta_k}(\boldsymbol{e}_{i,1}^d, \ldots, \boldsymbol{e}_{i,j}^d, \ldots, \boldsymbol{e}_{i,l_d}^d), \tag{2}$$

where $\boldsymbol{e}_{i,j}^d$ is the embedding of the $j$-th word in $d_i$ initialized using GloVe. $\{\boldsymbol{h}_{i,j}^d\}_{i=1,j=1}^{i=m,j=l_d}$ are fed to the knowledge processor to ground response prediction on $D$.

Different from Transformer Memory Network (Dinan et al., 2019), our model does not perform knowledge selection in the encoding phase (e.g., via attention over $\{\boldsymbol{h}_{i,j}^d\}_{i=1,j=1}^{i=m,j=l_d}$), but leaves it to the decoding phase. This could remove the dependency between context encoding and knowledge encoding, and facilitate us to estimate $\theta_e$ and $\theta_k$ with $\mathcal{D}_P$ and $\mathcal{D}_C$ respectively.

### 2.2.2 DISENTANGLED DECODER

The decoder maintains a hidden sequence $\{\boldsymbol{s}_t\}_{t=1}^{l_r}$. Let $\boldsymbol{e}_{t-1}^r$ be the embedding of the word predicted at step $t-1$, then $\boldsymbol{s}_t$ is defined by

$$\boldsymbol{s}_t = \text{GRU}_{\theta_d}(\boldsymbol{e}_{t-1}^r, \boldsymbol{s}_{t-1}), \tag{3}$$

where $\boldsymbol{s}_0 = \boldsymbol{h}_{l_u}^u$. Based on $\{\boldsymbol{s}_t\}_{t=1}^{l_r}$, the three components are defined as follows:

**Language Model.** The language model predicts a word based on $\boldsymbol{s}_t$. For words that do not need the context and the document (e.g., function words), employing the language model may enhance decoding speed without loss of accuracy. Formally, the generation probability is defined by

$$P(w_t^r|w_{1:t-1}^r) = \texttt{MLP}_{\theta_l}(\boldsymbol{s}_t). \tag{4}$$

**Context Processor.** The context processor predicts a word by attending over $\{\boldsymbol{h}_i^u\}_{i=1}^{l_u}$. The word could be either fetched from the vocabulary or copied from the context $U$. Let $\boldsymbol{c}_t^u$ be the context vector at step $t$, then $\boldsymbol{c}_t^u$ can be formulated as

$$\boldsymbol{c}_t^u = \sum_{i=1}^{l_u} \alpha_{t,i}\boldsymbol{h}_i^u, \tag{5}$$

where $\alpha_{t,i} = \exp(e_{t,i})/\sum_i \exp(e_{t,i})$ denotes the attention distribution and $e_{t,i} = g_{\theta_s}(\boldsymbol{s}_t, \boldsymbol{h}_i^u) = v^\top\tanh(W_h\boldsymbol{h}_i^u + W_s\boldsymbol{s}_t + b)$. The generation probability is defined by

$$P(w_t^r|U, w_{1:t-1}^r) = p_{\texttt{gen}}P_{\texttt{vocab}}(w_t^r|U, w_{1:t-1}^r) + (1 - p_{\texttt{gen}}) \sum_{i:w_i^u=w_t^r} \alpha_{t,i}. \tag{6}$$

In Equation (6), the first term models the correspondence between a context and a response, and is formulated as $P_{\texttt{vocab}}(w_t^r|U, w_{1:t-1}^r) = \texttt{MLP}_{\theta_v}([\boldsymbol{s}_t; \boldsymbol{c}_t^u])$. The second term models the copy mechanism, and $p_{\texttt{gen}} = \texttt{MLP}_{\theta_g}([\boldsymbol{c}_t^u; \boldsymbol{s}_t; \boldsymbol{e}_{t-1}^r]) \in [0, 1]$ a trade-off between the two terms.

**Knowledge Processor.** The knowledge processor goes through the document $D$ by a hierarchical attention mechanism, and predicts a word in a similar way as Equation (6). Formally, let $\{\beta_{t,i}^s\}_{i=1}^m$ and $\{\beta_{t,i,j}^w\}_{i=1,j=1}^{i=m,j=l_d}$ be the sentence-level attention distribution and the word-level attention distributions respectively at step $t$, then $\forall i \in \{1, \ldots, m\}$ and $\forall j \in \{1, \ldots, l_d\}$, $\beta_{t,i}^s$ and $\beta_{t,i,j}^w$ are calculated by

$$\beta_{t,i}^s = \exp(g_{\theta_{s'}}(\boldsymbol{s}_t, \hat{\boldsymbol{h}}_i^d))/\mathbf{Z}_s; \quad \beta_{t,i,j}^w = \exp(g_{\theta_{s'}}(\boldsymbol{s}_t, \boldsymbol{h}_{i,j}^d))/\mathbf{Z}_w, \tag{7}$$

where $\mathbf{Z}_s$ and $\mathbf{Z}_w$ are normalization factors, and $\hat{\boldsymbol{h}}_i^d$ represents the average pooling of $\{\boldsymbol{h}_{i,j}^d\}_{j=1}^{l_d}$. A knowledge vector $\boldsymbol{c}_t^d$ that is analogous to $\boldsymbol{c}_t^u$ is then defined by

$$\boldsymbol{c}_t^d = \sum_{i=1}^m \beta_{t,i}^s\hat{\boldsymbol{h}}_i^d. \tag{8}$$

Finally, the generation probability is formulated as

$$P(w_t^r|D, w_{1:t-1}^r) = p_{\texttt{gen}}'P_{\texttt{vocab}}(w_t^r|D, w_{1:t-1}^r) + (1 - p_{\texttt{gen}}') \sum_{i,j:w_{i,j}^d=w_t^r} \beta_{t,i,j}, \tag{9}$$

where $\beta_{t,i,j} = \beta_{t,i}^s \cdot \beta_{t,i,j}^w$, $w_{i,j}^d$ is the $j$-th word of $d_i$, $P_{\texttt{vocab}}(w_t^r|D, w_{1:t-1}^r) = \texttt{MLP}_{\theta_{v'}}([\boldsymbol{s}_t; \boldsymbol{c}_t^d])$, and $p_{\texttt{gen}}' = \texttt{MLP}_{\theta_{g'}}([\boldsymbol{c}_t^d; \boldsymbol{s}_t; \boldsymbol{e}_{t-1}^r])$ acts as a trade-off between the common term and the copy term.

### 2.2.3 Decoding Manager

The three components are controlled by the decoding manager with one picked up at each step of response prediction. Then, the probability to predict word $w_t^r$ can be formulated as

$$P(w_t^r|U, D, w_{1:t-1}^r) = [P(w_t^r|w_{1:t-1}^r); P(w_t^r|U, w_{1:t-1}^r); P(w_t^r|D, w_{1:t-1}^r)] \cdot \pi_t. \tag{10}$$

In training, to handle the discrete and undifferentiable process, we employ the Gumbel trick (Jang et al., 2016) and define $\pi_t$ as

$$\pi_t = \texttt{gumbel\_softmax}(f_\pi(\boldsymbol{s}_{t-1}), \tau) \in \mathbb{R}^{3\times 1}, \tag{11}$$

where $f_\pi(\cdot) = \texttt{MLP}_{\theta_\pi}(\cdot)$, $\texttt{gumbel\_softmax}(\cdot)$ denotes the Gumbel-Softmax function (Jang et al., 2016), and $\tau$ is the temperature (hyperparameter). $\pi_t$ approaches to a one-hot vector when $\tau \rightarrow 0$. We start from a high temperature and gradually reduce it. In test, we discretize $\pi_t$ as a one-hot vector according to the distribution in Equation (11).

## 2.3 LEARNING DETAILS

Let us denote $\{\theta_{ol}, \theta_{oc}, \theta_{od}\}$ as the parameters of word embedding in response prediction corresponding to the language model, the context processor, and the knowledge processor respectively. For simplicity, we let $\theta_{oc} = \theta_{od} = \theta_o$. Then $\{\theta_e; \theta_d; \theta_s; \theta_v; \theta_g; \theta_o\}$ (including parameters of the context encoder, parameters of the hidden states of the decoder, and parameters of the context processor) are estimated with maximum likelihood estimation (MLE) on $\mathcal{D}_C = \{(U_j^C, r_j^C)\}_{j=1}^M$.

To estimate $\theta_l$ (i.e., parameters of the language model) and $\theta_{ol}$, we construct a corpus $\mathcal{D}_{LM} = \{u_j^{LM}\}_{j=1}^{M'}$ with $u_j^{LM}$ a response or an utterance from a context in $\mathcal{D}_C$, and then learn the parameters with MLE on $\mathcal{D}_{LM}$ with $\theta_d$ fixed.

Inspired by Peters et al. (2018), we estimate $\theta_k$ (i.e., parameters of the knowledge encoder) using a bidirectional language model by minimizing the following loss function on $\mathcal{D}_P$:

$$\ell = -\frac{1}{N} \sum_{i=1}^N \Big( \sum_{t=1}^{l_d} (\log p(w_t|w_{1:t-1}) + \log p(w_t|w_{t+1:l_d}))\Big). \tag{12}$$

The remaining parameters $\{\theta_{s'}; \theta_{v'}; \theta_{g'}; \theta_\pi\}$ (i.e., parameters of the knowledge processor and parameters of the decoding manager) are learned with MLE on $\mathcal{D}_S$ with all other parameters fixed. Note that parameters of word embedding in the encoders are supposed to be included in $\theta_e$ and $\theta_k$.

**Remarks.** We focus on document-grounded dialogue generation in this work, but the approach proposed actually provides a recipe for a general solution to low-resource knowledge-grounded dialogue generation in which the knowledge could be a structured knowledge base, images, or videos. To do that, one only needs to modify the knowledge encoder and the knowledge processor to make them compatible with the specific type of knowledge, and pre-train the knowledge encoder, if possible, on single-modal knowledge data.

## 3 EXPERIMENTS

We test the proposed model on Wizard of Wikipedia (Wizard) published in Dinan et al. (2019) and CMU Document Grounded Conversations (CMU_DoG) published in Zhou et al. (2018b).

### 3.1 DATASETS AND EVALUATION METRICS

Both Wizard and CMU_DoG consist of open domain dialogues grounded on wiki articles, and the dialogues are collected from crowd-workers on Amazon Mechanical Turk. In Wizard, the articles cover a wide range of topics (totally $1,365$) such as bowling, Gouda cheese, and Arnold Schwarzenegger, etc. Each conversation happens between a wizard who has access to knowledge about a specific topic and an apprentice who is just eager to learn from the wizard about the topic. On average, each wizard turn is associated with $60.8$ sentences retrieved from the wiki articles and each sentence contains $30.7$ words. The data is split as a training set, a validation set, and a test set by the data owner. The test set is split into two subsets: Test Seen and Test Unseen. Test Seen contains new dialogues with topics appearing in the training set, while topics in Test Unseen never appear in the training set and the validation set, and thus the data allow us to examine the generalization ability of models. The task is to generate a response for each wizard turn based on the dialogue history and the retrieved knowledge. As pre-processing, for each wizard turn in the training/validation/test sets, the latest 128 words in the dialogue history are kept as a context. The pre-processing strictly follows the procedure in Dinan et al. (2019), and is conducted with the code published on ParlAI[1].

Different from Wizard, CMU_DoG focuses on movie domain (although covering various genres). In addition to wizard & apprentice, the data also contain dialogues between two workers who know the document and try to discuss the content in depth. Each document consists of $4$ sections and these sections are shown to the workers one by one every 3 turns (the first section lasts 6 turns due to initial greetings). On average, each section contains $8.22$ sentences and $27.86$ words per sentence. The data has been divided into a training set, a validation set, and a test set by the data owner. The task is

---

[1] https://github.com/facebookresearch/ParlAI/blob/master/projects/wizard_of_wikipedia

| Metrics
Models | PPL | F1 | BLEU-1 | BLEU-2 | BLEU-3 | BLEU-4 | Average | Extrema | Greedy |
|---|---|---|---|---|---|---|---|---|---|
| TMN (Dinan et al., 2019) | 66.5 | 15.9 | 0.184 | 0.073 | 0.033 | 0.017 | 0.844 | 0.427 | 0.658 |
| ITDD (Li et al., 2019) | 17.8 | 16.2 | 0.158 | 0.071 | 0.040 | 0.025 | 0.841 | 0.425 | 0.654 |
| FULL DATA | 23.0 | 18.0 | 0.218 | 0.115 | 0.075 | 0.055 | 0.835 | 0.434 | 0.658 |
| 1/2 DATA | 25.3 | 17.5 | 0.217 | 0.113 | 0.073 | 0.053 | 0.833 | 0.431 | 0.657 |
| 1/4 DATA | 29.2 | 16.9 | 0.212 | 0.105 | 0.064 | 0.044 | 0.833 | 0.429 | 0.658 |
| 1/8 DATA | 33.5 | 16.3 | 0.206 | 0.098 | 0.059 | 0.039 | 0.832 | 0.425 | 0.658 |
| 1/16 DATA | 38.6 | 15.7 | 0.197 | 0.091 | 0.052 | 0.033 | 0.834 | 0.428 | 0.655 |

Table 1: Evaluation results on Test Seen of Wizard.

to generate a response for each turn from a worker who has access to the document based on the dialogue history and the associated section as knowledge. Similar to Wizard, the latest 128 words in the dialogue history are kept as a context. More details of the datasets can be found in Appendix A.

We choose Reddit Conversation Corpus[2] cleaned by Dziri et al. (2018) as $\mathcal{D}_C$. The data contain $15,120,136$ context-response pairs for training and $830,777$ context-response pairs for validation. On average, each context consists of 3.5 utterances. We use the Wikipedia dump published on ParlAI[3] as $\mathcal{D}_P$. The training set and the validation set contain $5,233,799$ articles and $52,867$ articles respectively with the first paragraph kept for learning. Articles that appear in Wizard and CMU_DoG are removed beforehand. For both Wizard and CMU_DoG, the vocabulary is made up of top $60,000$ most frequent words appearing in $\mathcal{D}_S \cup \mathcal{D}_P \cup \mathcal{D}_C$ with other words regarded as $\langle \text{unk} \rangle$.

Following the common practice in evaluating open domain dialogue generation, we choose perplexity (PPL) of the ground-truth response, BLEU (Papineni et al., 2002), and BOW Embedding (Liu et al., 2016) as metrics. Besides, we also follow Dinan et al. (2019) and employ unigram F1 as a metric. BLEU and Embedding-based metrics are computed with an NLG evaluation open source available at `https://github.com/Maluuba/nlg-eval`, and unigram F1 is calculated with the code published at `https://github.com/facebookresearch/ParlAI/blob/master/parlai/core/metrics.py`. Besides quantitative evaluation, we also recruit human annotators to do qualitative analysis on response quality, which is presented in Appendix C.

### 3.2 BASELINES

The following models are selected as baselines:

**Transformer Memory Network (TMN).** The model proposed by Dinan et al. (2019) along with the release of the Wizard data. It is built upon a transformer architecture with an external memory hosting the knowledge. We implement the model using the code shared at `https://github.com/facebookresearch/ParlAI/blob/master/projects/wizard_of_wikipedia`.

**Incremental Transformer with Deliberation Decoder (ITDD).** A transformer-based model published very recently on ACL'19 (Li et al., 2019). The encoder incrementally represents multi-turn dialogues and knowledge, and the decoder conducts response decoding in two passes similar to the deliberation network in machine translation. We implement the model using the code shared at `https://github.com/lizekang/ITDD`.

Note that to make the comparison fair, we employ the end-to-end version of TMN without the knowledge regularization in learning. After all, one can include ground-truth signals on knowledge selection in both our model and TMN, and improve the two in the same way, although such signals are not available in most scenarios (e.g., in CMU_DoG).

### 3.3 EVALUATION RESULTS

To simulate a low-resource scenario, we start from using the full training data as $\mathcal{D}_S$, and gradually reduce the number of training examples by halving the training set. Note that baseline models are learned with the full training sets. Table 1 and Table 2 report evaluation results on Test Seen and Test Unseen of Wizard respectively, and Table 3 reports evaluation results on CMU_DoG. Through pre-training 95% parameters with the ungrounded dialogues and the plain text and fixing the parameters afterwards, our model holds the state-of-the-art performance in terms of most metrics on all test sets even when the training sets have been cut to $1/8$, and has stable performance on Test Unseen with

---

[2] `https://github.com/nouhadziri/THRED`
[3] `https://github.com/facebookresearch/ParlAI/tree/master/parlai/tasks/wikipedia`

| Metrics / Models | PPL | F1 | BLEU-1 | BLEU-2 | BLEU-3 | BLEU-4 | Average | Extrema | Greedy |
|---|---|---|---|---|---|---|---|---|---|
| TMN (Dinan et al., 2019) | 103.6 | 14.3 | 0.168 | 0.057 | 0.022 | 0.009 | 0.839 | 0.408 | 0.645 |
| ITDD (Li et al., 2019) | 44.8 | 11.4 | 0.134 | 0.047 | 0.021 | 0.011 | 0.826 | 0.364 | 0.624 |
| FULL DATA | 25.6 | 16.5 | 0.207 | 0.101 | 0.062 | 0.043 | 0.828 | 0.422 | 0.628 |
| 1/2 DATA | 27.7 | 16.7 | 0.208 | 0.103 | 0.064 | 0.045 | 0.827 | 0.421 | 0.647 |
| 1/4 DATA | 32.4 | 16.2 | 0.205 | 0.098 | 0.060 | 0.041 | 0.828 | 0.423 | 0.650 |
| 1/8 DATA | 35.8 | 16.0 | 0.201 | 0.093 | 0.054 | 0.035 | 0.831 | 0.419 | 0.653 |
| 1/16 DATA | 41.0 | 15.3 | 0.191 | 0.087 | 0.050 | 0.032 | 0.832 | 0.424 | 0.652 |

Table 2: Evaluation results on Test Unseen of Wizard.

| Metrics / Models | PPL | F1 | BLEU-1 | BLEU-2 | BLEU-3 | BLEU-4 | Average | Extrema | Greedy |
|---|---|---|---|---|---|---|---|---|---|
| TMN (Dinan et al., 2019) | 75.2 | 9.9 | 0.115 | 0.040 | 0.016 | 0.007 | 0.789 | 0.399 | 0.615 |
| ITDD (Li et al., 2019) | 26.0 | 10.4 | 0.095 | 0.036 | 0.017 | 0.009 | 0.748 | 0.390 | 0.587 |
| FULL DATA | 54.4 | 10.7 | 0.150 | 0.057 | 0.025 | 0.012 | 0.809 | 0.413 | 0.633 |
| 1/2 DATA | 57.0 | 10.4 | 0.142 | 0.052 | 0.022 | 0.010 | 0.808 | 0.414 | 0.635 |
| 1/4 DATA | 61.7 | 10.5 | 0.131 | 0.046 | 0.019 | 0.009 | 0.781 | 0.402 | 0.613 |
| 1/8 DATA | 67.6 | 10.2 | 0.121 | 0.044 | 0.019 | 0.009 | 0.787 | 0.407 | 0.622 |

Table 3: Evaluation results on CMU_DoG.

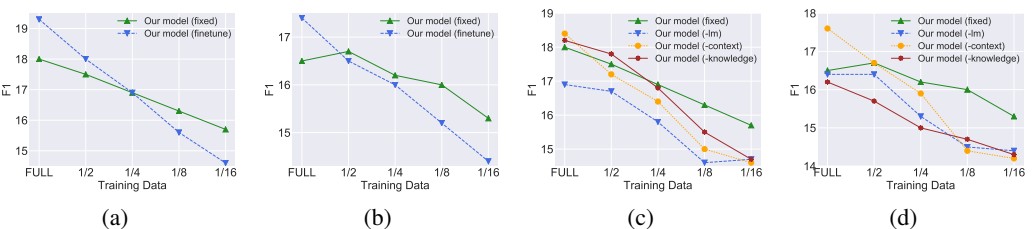

(a)    (b)    (c)    (d)

Figure 2: Performance of variants of the proposed model on Wizard. (a) Comparison of parameter fine-tuning and parameter fixing on Test Seen. (b) Comparison of parameter fine-tuning and parameter fixing on Test Unseen. (c) Results of pre-training ablation on Test Seen. (d) Results of pre-training ablation on Test Unseen.

respect to different training sizes. Particularly, the model achieves more significant improvement over the baselines on Test Unseen, and when the training set shrinks, the performance gap on Test Seen and Test Unseen becomes marginal. The results show a good generalization ability of the proposed model on out-of-domain knowledge. ITDD achieves low PPL on both Test Seen and CMU_DoG, which may stem from overfitting by the two-pass decoder. As an evidence, the model is just comparable with TMN on most metrics except PPL on Test Seen and CMU_DoG, and is worse than our model on Test Unseen even in terms of PPL.

## 3.4 DISCUSSIONS

In addition to the performance of the model under low-resource settings, we are also curious about **Q1:** what if we fine-tune the pre-trained parameters, rather than fixing them, with the training data of the knowledge-grounded dialogues, given that pre-training → fine-tuning has become the fashion in NLP research and engineering? **Q2:** can we somehow leverage the ungrounded dialogues and the plain text in learning of TMN, and in this case, will there be any change in the comparison with our model? and **Q3:** what is the impact of pre-training to different components of the proposed model?

**Answer to Q1:** Figure 2(a) and Figure 2(b) compare our models with fine-tuned parameters and fixed parameters on Test Seen and Test Unseen respectively. Basically, when there are enough training data (e.g., > 1/2), fine-tuning can further improve the model on both in-domain and out-of-domain knowledge. On the other hand, when the training size is small, which is the assumption of the paper, fine-tuning may cause overfitting and lead to performance drop on the test sets. Test Unseen is more vulnerable than Test Seen, and the smaller the training size is, the bigger the gap is between the model with fixed parameters and the model with fine-tuned parameters. Therefore, in a low-resource setting (e.g., less than 5k training dialogues), it is better to fix the pre-trained parameters and only estimate the remaining 5% parameters with the training data.

**Answer to Q2:** Normally, it is not trivial to learn an entangled architecture like TMN with ungrounded dialogues and plain text. However, to make the comparison even more fair, we first pre-train a transformer-based encoder-decoder with the Reddit data. The encoder is fixed and used for TMN, and

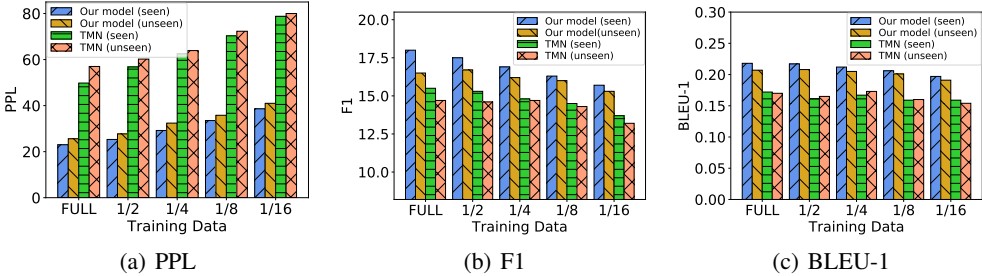

Figure 3: Comparison with pre-trained TMN on Wizard.

the parameters of the decoder is used to initialize the parameters of the decoder of TMN. Then, we pre-train the document representation in TMN with the Wikipedia dump. Finally, the knowledge attention in encoding and the decoder are learned (fine-tuned) with the training data of knowledge-grounded dialogues, as knowledge and dialogue contexts are entangled in the two modules. Figure 3 compares the pre-trained TMN with our model. Even though we have tried our best to make TMN use $\mathcal{D}_C$ and $\mathcal{D}_P$, it is still much worse than our model. The results indicate the importance of disentangling to leveraging ungrounded dialogues and plain text for low-resource knowlededg-grounded dialogue generation.

**Answer to Q3:** Figure 2(c) and Figure 2(d) show the results of ablation study in terms of pre-training. *-lm* means that $\theta_l$ and $\theta_{ol}$ are estimated using $\mathcal{D}_S$ together with $\{\theta_{s'}; \theta_{v'}; \theta_{g'}; \theta_\pi\}$. Similarly, *-context* and *-knowledge* mean that pre-training is removed from $\{\theta_e; \theta_d; \theta_s; \theta_v; \theta_g; \theta_o\}$ and $\theta_k$ respectively. We can conclude that (1) pre-training is crucial to low-resource knowledge-grounded dialogue generation, since removing any component from pre-training causes performance drop when training data is small; and (2) in terms of impact to performance, *lm>context>knowledge* on Test Seen, while *knowledge>lm>context* on Test Unseen.

## 4 RELATED WORK

Research on end-to-end open domain dialogue generation is encouraged by the success of neural sequence-to-sequence models on machine translation (Sutskever et al., 2014). On top of the basic architecture (Shang et al., 2015; Vinyals & Le, 2015), various extensions have been made to tackle the safe response problem (Li et al., 2015; Xing et al., 2017; Zhao et al., 2017; Song et al., 2018; Tao et al., 2018; Qiu et al., 2019); to model dialogue history for multi-turn conversation (Serban et al., 2016; 2017); and to learn with advanced machine learning techniques (Li et al., 2016; 2017). Very recently, grounding response generation on a specific type of knowledge, such as triples from a knowledge base (Zhou et al., 2018a), documents (Ghazvininejad et al., 2018; Zhao et al., 2019), personas (Zhang et al., 2018), and images (Mostafazadeh et al., 2017), has emerged as a new fashion in the research of open domain dialogue systems. This work aligns with the trend by considering document-grounded dialogue generation. Our model is built upon state-of-the-art neural generation techniques such as attention (Bahdanau et al., 2015; Yang et al., 2016) and copying (See et al., 2017; Raghu et al., 2019; Yavuz et al., 2019), but is unique in that components are pre-trained from various sources, thanks to the disentangled design. Thus, rather than testing new architectures on the benchmarks, our main contribution lies in investigation of knowledge-grounded dialogue generation under a low-resource setting with pre-training techniques, which roots in the requirement from practice.

The idea of "disentangling response decoding" is inspired by the similar research in representation learning that aims to seek a representation axis aligning with the generative factors of data (Bengio et al., 2013). State-of-the-art models are built within the framework of variational auto-encoding (Kingma & Welling, 2013) either under an unsupervised assumption (Higgins et al., 2017; Kim & Mnih, 2018; Chen et al., 2016; 2018) or aided by a few labels (Narayanaswamy et al., 2017; Locatello et al., 2019). In this work, we borrow the concept of "disentangling", but apply it to the structure of the decoder of a response generation model. The result is a few independent components that allow asynchronous parameter estimation. The work is also encouraged by the recent breakthrough on pre-training for NLP tasks (Peters et al., 2018; Devlin et al., 2018; Yang et al., 2019; Liu et al., 2019; Song et al., 2019). We take advantage of disentanglement, and employ pre-training techniques to tackle the low-resource challenge in the task of knowledge-grounded dialogue generation.

## 5 CONCLUSIONS

We study knowledge-grounded dialogue generation under a low-resource setting. To overcome the challenge from insufficient training data, we propose decomposing the response decoder into independent components in which most parameters do not rely on the training data any more and can be estimated from large scale ungrounded dialogues and unstructured documents. Evaluation results on two benchmarks indicate that our model achieves the state-of-the-art performance with only $1/8$ training data, and exhibits a good generalization ability on out-of-domain knowledge.

### ACKNOWLEDGMENTS

We would like to thank the reviewers for their constructive comments. This work was supported by the National Key Research and Development Program of China (No. 2017YFC0804001), the National Science Foundation of China (NSFC No. 61876196 and NSFC No. 61672058). Rui Yan was sponsored as the young fellow of Beijing Academy of Artificial Intelligence (BAAI). Rui Yan is the corresponding author.

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

# APPENDIX

## A  DETAILS OF DATASETS

Table 4 reports the statistics of the Wizard data and the CMU_DOG data.

| | Wizard of Wikipedia | | | | CMU_DoG | | |
|---|---|---|---|---|---|---|---|
| | Train | Valid | Test Seen | Test Unseen | Train | Valid | Test |
| Number of Utterances | 166,787 | 17,715 | 8,715 | 8,782 | 74,717 | 4,993 | 13,646 |
| Number of Conversations | 18,430 | 1,948 | 965 | 968 | 3,373 | 229 | 619 |
| Number of Topics/Documents | 1,247 | 599 | 533 | 58 | 30 | 30 | 30 |
| Average Turns per Dialogue | 9.0 | 9.1 | 9.0 | 9.1 | 22.2 | 21.8 | 22.0 |

Table 4: Statistics of the two datasets.

## B  MORE IMPLEMENTATION DETAILS

In both Wizard and CMU_DOG, we set the size of word embedding as $300$, the hidden size of the context encoder, the knowledge encoder, and the decoder as $1024$. The context encoder and the decoder have 3 layers respectively. The $g_{\theta_s}$ and $g_{\theta_{s'}}$ are similarity functions which contain two single-layer feed-forward networks (FFNs) of size $512$ with tanh non-linearity. The $\texttt{MLP}_{\theta_l}$, $\texttt{MLP}_{\theta_v}$ and $\texttt{MLP}_{\theta_v}$ are two-layer FFNs of size $1024$ and $300$ respectively. The $\texttt{MLP}_{\theta_g}$, $\texttt{MLP}_{\theta_{g'}}$ and $\texttt{MLP}_{\theta_\pi}$ are single-layer FFNs. All models are learned with Adam (Kingma & Ba, 2015) optimizer with $\beta_1 = 0.9$, $\beta_2 = 0.999$, and an initial learning rate $= 5e-4$. We increase the learning rate linearly for the first 5000 training steps and decrease it thereafter proportionally to the inverse square root of the step number. We set the initial temperature, the minimum temperature, and the anneal rate of $\texttt{gumbel\_softmax}$ as $1.0$, $0.6$, and $4e-5$ respectively. In training, we choose $64$ as the size of mini-batches, and add dropout to $g_{\theta_{s'}}$ and $\texttt{MLP}_{\theta_{v'}}$, but do not see much difference. Early stopping on validation is adopted as a regularization strategy. We employ beam search in response decoding with a beam size 5. We add weak supervision to guide the training of the decoding manager where the words that belong to modal verbs[4] are forced to be classified as language model.

## C  HUMAN EVALUATION

| Metrics
Models | Seen | | | | Unseen | | | |
|---|---|---|---|---|---|---|---|---|
| | Fluency | Context
Coherence | Knowledge
Relevance | Kappa | Fluency | Context
Coherence | Knowledge
Relevance | Kappa |
| TMN (Dinan et al., 2019) | 1.26 | 0.51 | 0.47 | 0.60 | 1.40 | 0.35 | 0.46 | 0.68 |
| ITDD (Li et al., 2019) | 1.69 | 1.18 | 1.16 | 0.70 | 1.72 | 0.73 | 0.71 | 0.69 |
| 1/4 DATA | 1.77 | 1.54 | 1.17 | 0.58 | 1.75 | 1.26 | 1.18 | 0.57 |
| 1/8 DATA | 1.68 | 1.44 | 1.13 | 0.60 | 1.73 | 1.21 | 1.25 | 0.57 |

Table 5: Human evaluation results on Wizard.

The goal of human study is to get more insights on quality of responses generated by different models from human annotators. To this end, we randomly sample 300 examples from Test Seen and Test Unseen respectively, and recruit 3 well educated native speakers as the annotators. Comparison is conducted among TMN, ITDD, our model (with $1/4$ training data), and our model (with $1/8$ training data). On each test set, for each of the 300 examples, an annotator is provided with a context, the ground-truth knowledge, and responses provided by the models under evaluation (the top one response in beam search). Responses are pooled and randomly shuffled to hide their sources. Then, each annotator judges the responses from three aspects including *fluency*, *context coherence*, and *knowledge relevance*, and assigns a score from $\{0, 1, 2\}$ to each of the response on each aspect, in which 0 means bad, 1 means fair, and 2 means good. Each response receives 3 scores on each aspect, and agreement among the annotators are calculated with Fleiss' kappa (Fleiss, 1971). Table 5 shows the average scores on the three aspects. Overall, the proposed model achieves the state-of-the-art performance in terms of all the three aspects on both Test Seen and Test Unseen when only

---

[4]"can", "would", "could", "will", "should", "may"

$1/8$ training examples are left. All kappa values exceed or are close to $0.6$, indicating substantial agreement among the annotators. The results are consistent with those reported in Table 1 and Table 2. Our model estimates the decoder with abundant extra resources, and ITDD exploits a two-pass decoder. Therefore, both of the two models can provide grammatical and fluent responses, no matter the background knowledge is within the domain of training or out of the domain of training. On the other hand, with the $15M$ Reddit data in learning of the context processor, our model can make the dialogues more coherent than the baselines, although there is a little drop on Test Unseen compared to Test Seen. Since the model only obtains limited guidance from training in terms of the connection between the knowledge and the dialogues, how to make the responses relevant to the knowledge is still challenging, although our model has done a better job than the baselines.

| [Knowledge] | ✧ the lineup , when first signed to geffen records in 1986 , consisted of vocalist axl rose , lead guitarist slash , rhythm guitarist izzy stradlin , bassist duff mckagan , and drummer steven adler . |
| | ✧ guns n ' roses has **released six studio albums** , accumulating sales of more than 100 million records worldwide , including 45 million in the united states , making them the 41st best-selling artist of all time . |
| | ✧ guns n ' roses ' debut album , `` appetite for destruction '' ( 1987 ) , reached number one on the `` billboard '' 200 a year after its release , on the strength of `` sweet child o ' mine '' , the band 's only single to reach number one on the `` billboard '' hot 100 . |
| | ✧ guns n ' roses , often abbreviated as gnr , is an **american hard rock band from los angeles** , california , formed in 1985 . |
| | ✧ the album has sold approximately 30 million copies worldwide , including 18 million units in the united states , making it the best-selling debut album of all time in the us , as well as the eleventh best-selling album in the united states . |
| | ✧ the success of the debut was followed by the eight-song album `` g n ' r lies '' ( 1988 ) which reached number two on the `` billboard '' 200 . |
| | ✧ the twin albums `` use your illusion i '' and `` use your illusion ii '' ( 1991 ) debuted at number two and number one on the `` billboard '' 200 respectively and have sold a combined 35 million copies worldwide , including 14 million units in the united states . |
| [Context] | have you heard of the band guns n ' roses ? they are my favorite band ever . |
| [Groundtruth] | sure the band was formed inn 1985 in los angeles , california . what is your favorite album ? |
| [ TMN ] | they are a great band . |
| [ITDD] | i have n't , but i do know that they are the most popular band in the us . |
| Our Model | me too ! they 're an american rock band from los angeles and has released six studio albums . |

Generated by Language model    Generated by Context Processor    Generated by Knowledge Processor

Table 6: A case from Test Unseen of Wizard.

Table 6 shows an example from Test Unseen, from which we can see that the response from our model (with $1/8$ training data) not only smoothly catches the context, but also expands the topic with proper pieces of knowledge (highlighted in red). On the other hand, responses from the baselines just reply to the context but lose the connection with the knowledge, as we have analyzed with the results in Table 5. Moreover, we also visualize the sources of words in the response with colors. Basically, words that have weak or no correlation with the context and the knowledge are generated by the language model, words that connect with the context but have nothing to do with the knowledge are generated by the context processor, and words that are copied from the knowledge are generated by the knowledge processor.

# D    COMPARISON WITH MASS

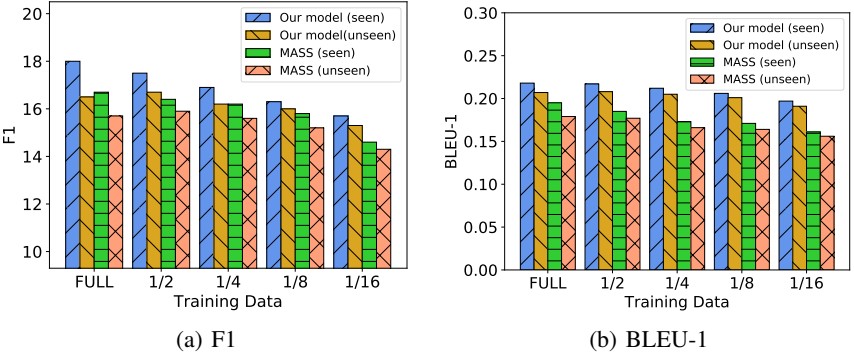

(a) F1                    (b) BLEU-1

Figure 4: Comparison with MASS on Wizard.

We compare our model with MASS (Song et al., 2019), a pre-training technique that achieves state-of-the-art performance on several language generation tasks such as machine translation, text summarization, and conversational response generation. MASS firstly pre-trains an encoder-decoder architecture with large-scale monolingual data from WMT News Crawl datasets by reconstructing a fragment of a sentence from the remaining, and then fine-tunes the architecture on downstream language generation tasks. We use the code and the model published at `https://github.com/microsoft/MASS`. The original model is for sequence-to-sequence generation. To adapt it to the knowledge-grounded dialogue generation task, we concatenate the knowledge sentences and conversational history as a long context as the input of the encoder.

Figure 4 shows the evaluation results. Note that we do not include PPL as a metric like in Figure 3, since MASS performs generation with sub-words, and thus is not comparable with our model on PPL. On both Test Seen and Test Unseen, our model consistently outperforms MASS over all training sizes. The reason might be that "mask then predict", which is basically the pre-training strategy exploited by MASS, is not an effective way to leverage the text data for knowledge-grounded dialogue generation, since the task needs more complicated operations such as deep copying. Another reason might be that MASS is designed for the sequence-to-sequence generation task and isn't compatible with the knowledge-grounded response generation task which has extra knowledge input.

# E   ABLATION OVER COMPONENTS

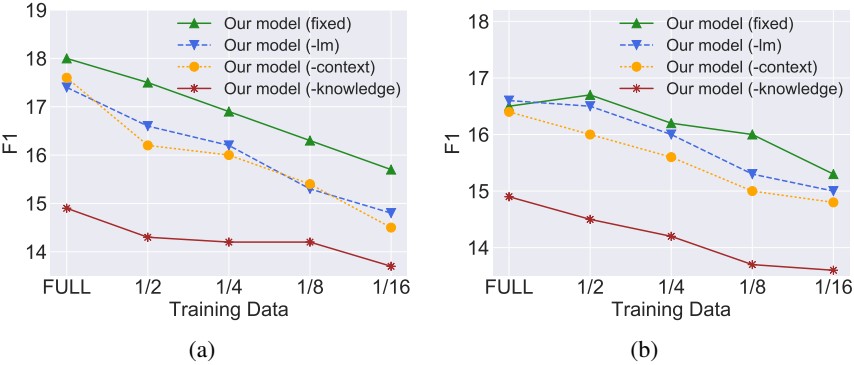

(a)                                    (b)

Figure 5: Ablation study over the three components of the decoder. (a) Results on Test Seen. (b) Results on Test Unseen.

We conduct ablation study over the language model, the context processor, and the knowledge processor by completely dropping any of them from the decoding manager (in both training and test). Figure 5(a) and Figure 5(b) report the results on Test Seen and Test Unseen respectively. First of all, all the three components are useful, since removing any of them in general will cause performance drop. Second, in terms of importance, knowledge processor>context processor>language model. The explanation is that (1) part of the function of the language model may be covered by the context processor and the knowledge processor after it is removed[5], since both the context processor and the knowledge processor also contain language models, although in the full model, the language model generates 17% words in the responses of Test Seen and Test Unseen; (2) the context processor is important (generating 27% words), but not always, since a large proportion of responses in the Wizard data highly depend on the knowledge (e.g., the examples shown in (Dinan et al., 2019)); (3) the knowledge processor (generating 56% words) is the most important component due to the nature of the Wizard data. The results also remind us that perhaps we can try pre-training the language model with larger and more heterogeneous data such as Common Crawl in the future.

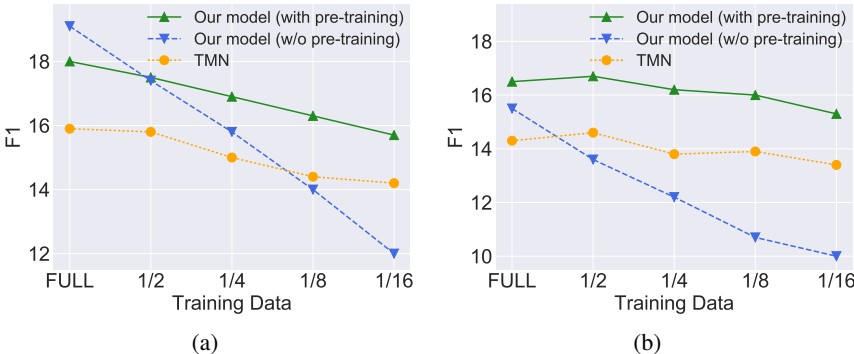

Figure 6: Comparison with the proposed model without pre-training. (a) Results on Test Seen. (b) Results on Test Unseen.

## F    COMPARISON WITH NON-PRETRAINING

Figure 6(a) and Figure 6(b) compare two versions of our model on Test Seen and Test Unseen respectively. One version is the model pre-trained using ungrounded dialogues and documents, and the other version is the one trained with knowledge-grounded dialogues (i.e., no pre-training is performed). Besides, we also include the results of TMN to get more insights. We can see that when there are enough training data (e.g., full data), our model without pre-training outperforms both TMN and the pre-trained version on Test Seen. This is because the attention and copying operations can well capture the correlation among the knowledge, the contexts, and the responses in the training data, while in the pre-trained version, only a small proportion of the model can benefit from the training data, and a large proportion may suffer from the gap between the knowledge-grounded dialogues collected from crowd-sourcing and the ungrounded dialogues and documents collected from the Web. However, when the training size shrinks, which is basically the problem we study in the paper, the performance of our model without pre-training drops dramatically, and becomes even worse than that of TMN on Test Seen when the training size is no more than $1/8$. This is because when training data is not enough, our model is more prone to overfit the small training set than TMN, and thus results in bad generalization ability. In the low-resource setting, pre-training, especially with the disentangled decoder if we consider the results in Figure 3, is an effective approach to obtaining good generalization ability on test data. The conclusions are further verified by the comparison on Test Unseen, where non-pre-training is worse than pre-training over all training sizes, and non-pre-training quickly drops below TMN when the training data is halved. On Test Unseen, with $1/8$ training data, the pre-trained model achieves the performance of the model learned from the full training data without pre-training.

---

[5]"Part of" is because the language model is pre-trained with monolingual Reddit data, which is different from the context processor and the knowledge processor.

