# OpenReview forum: "Low-Resource Knowledge-Grounded Dialogue Generation"
_ICLR.cc/2020/Conference — Accept (Poster)_

### Official Review · AnonReviewer2 · 2019-10-17
**Official Blind Review #2**

**Rating:** 8

**Review:**

This paper presents an encoder-decoder model architecture for knowledge-grounded dialogue generation in a low-resource setting. The model includes two GRU-based encoders which represents knowledge and dialogue context independently from each other. The decoder also has the following three independent components: language model, context processor, and document reader, each of which works as an individual response decoder. While the language model considers the decoder's hidden state only, the context processor and the document reader apply the copy mechanism from dialogue context and knowledge, respectively. Then, the decoding manager generates the final distributions by aggregating the component outputs with Gumbel softmax.

The main contribution of this work is the disentangled architecture where the following three types of internal components are trained separately on different data from each other. Firstly, context encoder, context processor and language model are trained on un-grounded dialogues with no knowledge. Secondly, knowledge encoder can be built purely on knowledge sources in parallel with the dialogue-based components. Finally, document reader and decoding manager require knowledge-grounded dialogues as training data.

The experimental results show that the proposed architecture help to resolve the low resource problems of knowledge-grounded dialogue generation. The model achieved state-of-the-art performances on both Wizard of Wikipedia and CMU DoG even with only 1/8 of the training data.

Overall, this is a well written paper with reasonable ideas supported by strong empirical evidences.

Please find below some questions and minor comments:
- Would it be possible to evaluate the performance of the model with no grounded dialogues? It would be interesting to see how well the model works only with the un-grounded Reddit and Wikipedia data.
- Have you thought about plugging in other language models trained on larger amount of texts such as Common Crawl? It seems possible, since the language model itself has no dependency to the context or the knowledge.
- What do you think of generalizing the architecture for general dialogue generation problems with no knowledge grounding? This disentangled decoder may be applied to the language model trained on the larger amount of texts and the context model on the smaller amount of dialogues.
- I suggest to provide some actual outputs from the models to show the advantage of the proposed method qualitatively.
- Could you possibly make the subfigures of Figure 2 and 3 larger? They are now too tiny to see the details.


**Experience Assessment:**

I have published in this field for several years.

**Review Assessment: Checking Correctness Of Derivations And Theory:**

I assessed the sensibility of the derivations and theory.

**Review Assessment: Checking Correctness Of Experiments:**

I carefully checked the experiments.

**Review Assessment: Thoroughness In Paper Reading:**

I read the paper at least twice and used my best judgement in assessing the paper.

---

> ### Author Response · Authors · 2019-11-15
> **Response to Reviewer #2**
>
> Thank you for the helpful comments!
>
> Q1: Would it be possible to evaluate the performance of the model with no grounded dialogues? It would be interesting to see how well the model works only with the un-grounded Reddit and Wikipedia data.
>
> A1: The reviewer describes another challenging setting, namely unsupervised learning. However, our current model can not handle this setting because we need a few training examples to train the parameters of the decoding manager and the parameters of the knowledge processor (i.e., a low-resource  setting). Otherwise, it is hard to know how to connect the knowledge with the conversation. Anyway, we recognize such an unsupervised setting is a valuable research direction and we leave it as future work.
>
>
> Q2: Have you thought about plugging in other language models trained on a larger amount of texts such as Common Crawl? It seems possible, since the language model itself has no dependency to the context or the knowledge.
>
> A2: Yes, you definitely point out a reasonable and useful direction for further improving the performance of our model. Due to the short interval of rebuttal, we would like to leave it as future work. Thank you for the suggestion.
>
>
> Q3: What do you think of generalizing the architecture for general dialogue generation problems with no knowledge grounding?
>
> A3: It is an interesting and forward-looking question! This may be useful especially when clean context-response pairs are not enough. However, this is not the main point of our paper but we are interested in trying it in the future.
> In addition, in very recently, pre-training has begun to attract attention in response generation, and a few papers emerging very recently have considered pre-training in dialogue generation such as \url{https://arxiv.org/pdf/1911.04700.pdf} and \url{https://arxiv.org/pdf/1911.00536.pdf}. We believe that pre-training will be an important direction for dialogue generation in the future.
>
>
> Q4: I suggest to provide some actual outputs from the models to show the advantage of the proposed method qualitatively.
>
> A4: For a clear illustration, we add a case study to show the advantage of the proposed model in the revised manuscript. Please refer to Table 6 in Appendix C for details.
>
>
> Q5: Could you possibly make the subfigures of Figure 2 and 3 larger? They are now too tiny to see the details.
>
> A5: We can do that with extra space (may go out of 8 pages). At the current stage, we are not sure if we can do that. But at least, we can enlarge the font size of the legends, which has been done in the revised manuscript.

---

### Official Review · AnonReviewer1 · 2019-10-23
**Official Blind Review #1**

**Rating:** 8

**Review:**

The authors propose a novel framework for training a knowledge-grounded dialogue model. They decouple the three main elements of such a model - a language model, a context processor, and a document reader - and as such can pretrain each component separately. They achieve state-of-the-art results on two benchmark datasets, and can additionally obtain near-state-of-the-art results while training on a fraction of the task data.

Overall I think it’s a strong paper with a good set of experiments, baselines, and considers multiple datasets.

Disentangling the model elements is a clever way to allow for more robust pre-training, and indeed yields favorable results. The authors show that just the pre-training aspect is not the root cause of their boost in performance, as a pre-trained baseline model fails to replicate their best results. The contribution is broadly applicable to other areas in which data collation is more difficult; the authors additionally do a good job of pointing out that their knowledge encoder is not limited to text but can also use other knowledge grounding including images, videos, or a knowledge-base. Finally, the authors detail thorough ablation studies for their models.

One major thing missing from their ablation in Figure 2 is a setting when *no* pretraining is used. That would be much more comparable to the setting used for TMN, since that had *no* pretraining available to it. Alternative, adding pretraining to the baselines would be another good way to do this, which would help disentangle how much the architecture is helpful over the pretraining.

Although the authors point out that a major advantage of their architecture is that we can separate the pretraining for each of the components, I would also be interested to see how they find the model doing if a single source of pretraining is used. I.e., only reddit pretrained weights for all 3 components, or only wikipedia etc. I definitely don’t think that holds back this paper, just think it would provide some evidence of the value gained utilizing the disparate sources.


**Experience Assessment:**

I have published in this field for several years.

**Review Assessment: Checking Correctness Of Derivations And Theory:**

I did not assess the derivations or theory.

**Review Assessment: Checking Correctness Of Experiments:**

I assessed the sensibility of the experiments.

**Review Assessment: Thoroughness In Paper Reading:**

I read the paper at least twice and used my best judgement in assessing the paper.

---

> ### Author Response · Authors · 2019-11-15
> **Response to Reviewer #1**
>
> Thank you for the helpful comments.
>
> Q1: One major thing missing from their ablation in Figure 2 is a setting when *no* pretraining is used. That would be much more comparable to the setting used for TMN, since that had *no* pretraining available to it. Alternative, adding pretraining to the baselines would be another good way to do this, which would help disentangle how much the architecture is helpful over the pretraining.
>
> A1: We add a comparison between pre-training and non-pre-training. We also carefully compare the models with TMN (the baseline) with a grid scan on training sizes. All results can be found in Appendix F in the revised manuscript. Actually, in Figure 3, we have somehow added pre-training to TMN and compared it with the proposed model. The conclusion is the proposed model, thanks to the disentangled design, is better than the pre-trained TMN.
>
>
> Q2: I would also be interested to see how they find the model doing if a single source of pretraining is used.
>
> A2: Thank you for the suggestion. We would investigate this problem in future work. However, pre-training all the three components with Wikipedia might be difficult, since it is hard to learn how to understand (e.g., attention and copying) conversation contexts from Wikipedia. Similarly, we also do not expect a good result from pre-training the knowledge encoder with Reddit, since the data distribution of Reddit is quite different from that of Wikipedia.  Anyway, we can try in our future work.

---

### Official Review · AnonReviewer3 · 2019-11-03
**Official Blind Review #3**

**Rating:** 6

**Review:**

This paper studies knowledge-grounded dialogue response generation in the low-resource setting. More precisely, it proposes a disentangled decoder consisting of three components: language model, context processor, and document reader. Disentangled decoder architecture provides a flexibility to train (or pre-train) different components on different data, making it convenient for low-resource setting. Overall, it is a sound idea for low-resource setting with generally positive experimental results, but limited in novelty in terms of the proposed architecture (similar to [1*, 2*]) and disentangling language and knowledge idea (similar to [3*]), lacks comparison with baselines for low-resource setting, and lacks discussion/reference of a few closely related works.

Here are some of my questions and concerns for the paper:

Great to have some ablations in terms of pre-training to see its effect on different components. However, it would be quite useful to also have ablations over components by completely dropping a component (like LM) while training decoding manager. It would also be useful to see some statistics/discussion on the effect of different components in the actual generation of responses. It might also be useful to include qualitative examples of the generated responses annotated with predictions of different components for each generated word.

In Figure-2 (c) and (d), some ablation results are reported for when pre-training is removed for each of the three components independently. It is interesting, though, to see that removing pre-training does not hurt (might even improve) the performance (esp. for Test Unseen) much for FULL training data case. Is there a particular reason for this observation? Also, it makes me curious how the proposed model would perform without pre-training any of the components. Would it already outperform the baselines discussed in the paper? If so, are these baselines strong enough (SOTA or close to SOTA) to help draw a meaningful conclusion from comparison with them? For example, how would fine-tuning a pre-trained MASS  [4*] perform and compare as a baseline? Can authors comment on this?

The proposed approach is very similar in architecture to [1*, 2*, 3*], which are not discussed/referenced in the paper. Except for pre-training, the only difference from [2] is that copying and generation distributions are softly combined into separate distribution each, independently. Inducing a single output distribution is done instead by deciding which source to use by an MLP layer on decoder state as in Eq. 11. So, I think the authors need to better isolate what the core contribution of this paper is: disentangled decoder or pre-training strategy? If it is the proposed disentangled decoder architecture, then authors should compare with similar architectures [2, 3] in the low-resource setting by initializing encoder and decoder from pre-trained weights on the same Reddit corpus. If it is the pre-training strategy, then it should be compared with various pre-training strategies for sequence generation (e.g., [4*]) proposed recently. For example, it would be useful to include a comparison with fine-tuning a pre-trained MASS  [4*] with the same amount of WoW training data (changing from 1/16 to 1/1).

Presentation of the paper can be improved by 1) changing the name of “document reader” (maybe to “knowledge processor” similar to context) as it essentially attends on the document rather than reading, 2) using abstraction in the technical section to help simplify notation and make it more interpretable.


REFERENCES:
[1*] Get To The Point: Summarization with Pointer-Generator Networks, See et al.
[2*] DeepCopy: Grounded Response Generation with Hierarchical Pointer Networks, Yavuz et al.
[3*] Disentangling Language and Knowledge in Task-Oriented Dialogs, Raghu et al.
[4*] MASS: Masked Sequence to Sequence Pre-training for Language Generation, Song et al.

**Experience Assessment:**

I have published one or two papers in this area.

**Review Assessment: Checking Correctness Of Derivations And Theory:**

N/A

**Review Assessment: Checking Correctness Of Experiments:**

I carefully checked the experiments.

**Review Assessment: Thoroughness In Paper Reading:**

I read the paper thoroughly.

---

> ### Author Response · Authors · 2019-11-15
> **Response to Reviewer #3**
>
> Thank you for the valuable comments which guide us on how to improve the work.  Below we attempt to address your concerns:
>
> Q1: It would be quite useful to also have ablations over components by completely dropping a component (like LM) while training decoding manager.
>
> A1: We add the extra ablation study in Appendix E in the revised manuscript. The conclusion is that (1) all the components are useful, since removing any of them will cause performance drop; 2) in terms of importance, knowledge processor > context processor > language model. Please refer to Appendix E for a more detailed analysis.
>
>
> Q2: It would also be useful to see some statistics/discussion on the effect of different components in the actual generation of responses.
>
> A2: In the full model, the language model, the context processor and knowledge processor generate 17%, 27% and 56% words in responses respectively. We add the results and discussion in Appendix E in our revised manuscript.
>
>
> Q3: It might also be useful to include qualitative examples of the generated responses annotated with predictions of different components for each generated word.
>
> A3: For a clear illustration, we add a case study from Test Unseen showing the generated responses annotated with predictions of different components for each generated word in the revised manuscript. Please refer to Table 6 in Appendix C for details.
>
>
> Q4: How would the proposed model perform without pre-training any of the components? Would it already outperform the baselines discussed in the paper?
>
> A4: We add the comparison between pre-training and non-pre-training in Appendix F in the revised manuscript. Basically, without any pre-training, our model is worse than TMN (the baseline) when the training size is no more than 1/8 on Test Seen and when the training size is no more than 1/2 on Test Unseen. On Test Seen, only with the full training data, non-pre-training is better than pre-training, and on Test Unseen, non-pre-training is always worse than pre-training.  Thus, under low-resource settings (the problem we study in the paper), without pre-training, the proposed model is worse than the selected baselines. Or, we can say pre-training is crucial to make the proposed model generalize well under low-resource settings and on unseen data.
>
>
> Q5: Removing pre-training does not hurt (might even improve) the performance (esp. for Test Unseen) much for FULL training data case. Is there a particular reason for this observation?
>
> A5: Since the model without pre-training outperforms the model with pre-training on Test Seen when full training data is available (results are shown in Figure 6 in Appendix F), it is easy to understand in Figure 2(c) why removing some part of pre-training will result in performance improvement. Indeed, when training data is enough, which is NOT the problem we study in the paper, directly learning from the training data is not a bad choice, as we have analyzed in Appendix F. It is noteworthy that results in Figure 2 (c) and (d) are not necessarily upper and lower bounded by pre-training and non-pre-training. Pre-training is a complicated mechanism, especially when it is performed over heterogeneous sources (Reddit and Wikipedia) and combined. That is why non-pre-training is worse than pre-training on Test Unseen under full data, but removing pre-training from the context processor leads to the best performance. Similarly, pre-training is worse than non-pre-training on Test Seen under full data, but removing pre-training from the language model makes things even worse.
>
>
> Q6: The authors need to better isolate what the core contribution of this paper is: disentangled decoder or pre-training strategy?
>
> A6: Our core contribution is investigation of knowledge-grounded dialogue generation under a low-resource setting with pre-training techniques. We have further clarified it in the last paragraph of Introduction in the revised manuscript. To echo the reviewer's comments, we compare the model with MASS and present the results in Appendix D. Basically, our model consistently outperforms MASS over all training sizes on both Test Seen and Test Unseen. Detailed analysis is also presented in Appendix D. The disentangled decoder, although not entirely new in terms of architecture, is crucial to the success of pre-training, as we have analyzed after the results in Figure 3 in the main text.
>
>
> Q7: The proposed approach is very similar in architecture to [1*, 2*, 3*], which are not discussed/referenced in the paper.
>
> A7: Thank you for leading us to the related papers. We cited them in Related Work, and highlight that the major difference lies in the pre-training ideas.
>
>
> Q8: Presentation of the paper can be improved.
>
> A8: Thank you! We made some changes following the suggestions.

---

### Author Response · Authors · 2019-11-15
**Summary of Submission Changes**

We appreciate all reviewers for their constructive feedback and comments for the improvement of the paper. We have updated the manuscript with changes as follows:

1. We add an example of generated responses annotated with predictions of different components for each generated word in Appendix C (as shown in Table 6).

2. We compare our model with MASS (a pre-training technique that achieves state-of-the-art performance on several language generation tasks) in Appendix D with both quantitative results (Figure 4) and analysis.

3. We conduct an ablation study by completely dropping any of the components from the decoding manager. Quantitative results and analysis are presented in Appendix E.

4. We compare our pre-trained model with the model without pre-training by scanning the training size from 1 to 1/16. TMN over grid scan is also included in the comparison for more insights. Quantitative results and analysis are presented in Appendix F.

5. We change ''document reader'' to ''knowledge processor'',  cite the papers recommended by Reviewer #3, and discuss relationship and difference with these work in Related Work.

---

### Decision · Program_Chairs · 2019-12-19

**Decision:**

Accept (Poster)

**Comment:**

The paper considers the problem of knowledge-grounded dialogue generation with low resources. The authors propose to disentangle the model into three components that can be trained on separate data, and achieve SOTA on three datasets.

The reviewers agree that this is a well-written paper with a good idea, and strong empirical results, and I happily recommend acceptance.